# Diagnostic Yield of TEE in Patients with Cryptogenic Stroke and TIA with Normal TTE: A Systematic Review and Meta-Analysis

**Shamik Shah** [1,*] , **Preeti Malik** [2] , **Urvish Patel** [2] , **Yunxia Wang** [3] and **Gary S. Gronseth** [3]

1. Department of Neurology, Stormont Vail Health, Topeka, KS 66604, USA
2. Department of Public Health, Icahn School of Medicine at Mount Sinai, New York, NY 10029, USA; pmalik.ma@gmail.com (P.M.); dr.urvish.patel@gmail.com (U.P.)
3. Department of Neurology, The University of Kansas Health System, Kansas City, KS 66160, USA; YWANG@kumc.edu (Y.W.); GGRONSETH@kumc.edu (G.S.G.)
* Correspondence: drshahshamik@gmail.com

**Abstract:** Introduction: The role of transesophageal echocardiography (TEE) in cryptogenic stroke and transient ischemic attack (TIA) with normal transthoracic echocardiography (TTE) remains controversial in the absence of definite guidelines. We aimed to perform a systematic review and meta-analysis to estimate an additional diagnostic yield and clinical impact of TEE in patients with cryptogenic stroke and TIA with normal TTE. Methods: We performed a systematic review of cohort studies on PubMed using the keywords 'cryptogenic stroke', cryptogenic TIA', 'TEE', and 'TTE' with matching MeSH terms. We included studies with patients who had cryptogenic stroke or TIA and had normal TTE findings, where the study intended to obtain TEE on all patients and reported all TEE abnormalities. The studies containing patients with atrial fibrillation were excluded. All studies were evaluated for internal and external validity. Inverse variance random effects models were used to calculate the effect size, the number needed to diagnose, and the 95% confidence interval. Results: We included 15 studies with 2054 patients and found LA/LAA/aortic thrombus, valvular vegetation, PFO-ASA, valvular abnormalities, and complex aortic plaques on TEE. Of these, 37.5% (29.7%–45.1%) of patients had additional cardiac findings on TEE. Management of 13.6% (8.1%–19.1%) of patients had changed after TEE evaluation. Based on current guidelines, it should change management in 4.1% (2.1%–6.2%) of patients and could potentially change management in 30.4% (21.9%–38.9%) of patients. Sensitivity analysis was also performed with only class II studies to increase internal validity, which showed additional cardiac findings in 38.4% (28.5%–48.3%), changed management in 20.2% (8.7%–31.8%), should change management in 4.7% (1.5%–7.9%), and could potentially change management in 30.4% (17.8%–43.0%) of patients. Conclusions: The diagnostic yield of TEE to find any additional cardiac findings in patients with cryptogenic stroke or TIA is not only high, but it can also change management for certain cardiac abnormalities. TTE in cryptogenic stroke or TIA may mitigate future risks by tailoring the management of these patients.

**Keywords:** cryptogenic stroke; transient ischemic attack (TIA); transesophageal echocardiography (TEE); transthoracic echocardiography (TTE); cerebrovascular disease

## 1. Introduction

Of the 690,000 ischemic strokes that occur in the United States every year, approximately 30% are of unknown cause, or cryptogenic even after a thorough diagnostic work-up. Possible mechanisms underlying cryptogenic stroke (CS) include but are not limited to occult paroxysmal atrial fibrillation (AF) and other atrial cardiopathies, paradoxical embolism through a patent foramen ovale (PFO), or sub-stenotic atherosclerosis [1]. Even though the majority of the potential mechanisms in the CS are attributed to embolic phenomena which are fundamentally associated with the cardiac origin, there is still no consensus

on a thorough cardiac work-up of stroke patients to identify the stroke mechanism and improve secondary prevention strategies [1,2]. Transient ischemic attacks (TIAs) are most commonly caused by the embolic or thrombotic consequences of atherothrombotic disease, which is similar to the underlying pathological mechanism for cardiovascular disease [3]. TIAs are a warning that the patient is at risk of further vascular events, not only a recurrent stroke, and are associated with poor outcomes [4]. TIA management mainly focuses on detailed a work-up with a physician's choice of echocardiography and pharmacological management.

Studies have given evidence that both transthoracic echocardiography (TTE) and transesophageal echocardiography (TEE) are useful in identifying a potential cardiac source in patients with CS and TIA [5]. The literature has also shown the superiority of TEE over TTE in identifying potential mechanisms. One study by Brujin et al. showed that TEE identified a potential cardiac source in 40% of patients with ischemic stroke, in which TTE was nonrevealing [6]. Even though the TEE is a gold standard screening method to identify potential embolic sources in the absence of any other etiology, its routine use in guiding evidence base changes in the clinical management of CS patients is still questionable [7–9]. Despite this uncertainty, several studies have reported that TEE diagnostic findings not only had a significant impact on the clinical management of CS patients [10–12], but also resulted in secondary stroke prevention in these patients [13,14].

The aim of the present systematic review and meta-analysis is to estimate an additional diagnostic yield of TEE and the impact of TEE diagnosis on the clinical management of patients with cryptogenic stroke and TIA with normal TTE.

## 2. Materials and Methods

### 2.1. Endpoints

The primary aim of this systematic review and meta-analysis of published literature is to estimate the additional diagnostic yield of TEE in evaluating cardiac abnormalities over TTE. Additional diagnostic yield was evaluated by identifying TTE findings like left atrial/left atrial appendage thrombus, valvular vegetation, and abnormalities such as intracardiac tumor, PFO, and atrial septal abnormality (ASA). The secondary aim was to evaluate the impact of TEE diagnosis on clinical management in patients with CS and TIA with normal TTE. The secondary aim was evaluated by the number needed to diagnose one additional abnormality with TEE that can change management per current guidelines.

### 2.2. Search Strategy and Selection Criteria

This systematic review and meta-analysis followed the recommendation of the Preferred Reporting Items for Systematic Reviews and Meta-ANALYSES (PRISMA) statement [15] and was written according to the Meta-analysis of Observational Studies in Epidemiology (MOOSE) proposal [16].

A comprehensive search of the literature was conducted using PubMed and the following keywords: "cryptogenic stroke", "stroke", "transient ischemic attack", and "transesophageal echocardiography" and their matching MeSH terms from inception to December 2019. A flow diagram of the literature search and study selection process is described in Figure 1.

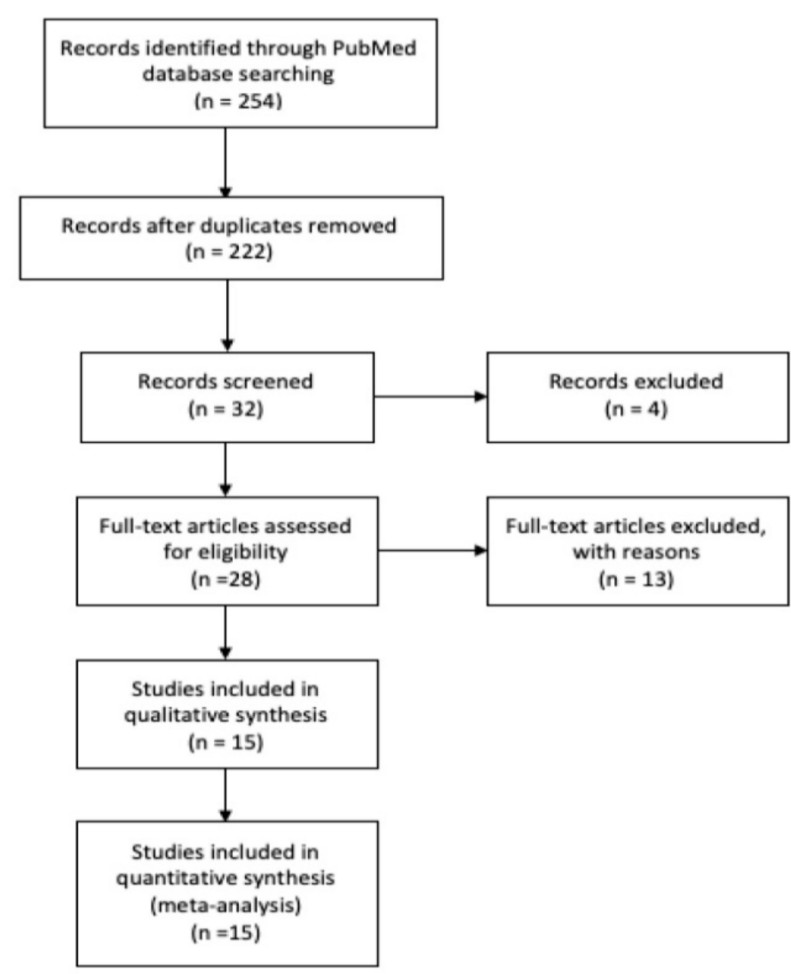

**Figure 1.** Flow diagram of literature search and study selection process.

*2.3. Inclusion Criteria*

- All prospective or retrospective observational cohort studies must include patients with cryptogenic stroke and/or TIA and a completed routine stroke work-up.
- All patients must have normal TTE results prior to TEE.
- All patients must have completed TEE and studies must have reported all TEE cardiac findings.

*2.4. Exclusion Criteria*

- Non-observational studies, non-English literature, non-full text, and animal studies were excluded.
- Patients with evidence of atrial fibrillation and/or heart disease and/or a history of anticoagulation prior to TEE were excluded.
- Studies with missing TEE cardiac findings were excluded.

*2.5. Study Selection*

Abstracts were reviewed, and articles were retrieved that met the inclusion criteria. All the retrieved studies and their reference to identify studies that may have been missed by the database search were scanned independently by three authors (S.S., U.P., P.M.). Any disagreement was resolved through consensus. Studies were excluded from further evaluation if they had no documented routine stroke work-up, no TTE performed or data

available, did not report all cardiac TEE findings, included patients with AF, or did not have a separate analysis for patients without AF.

### 2.6. Data Collection

From the included studies, we extracted the variables LA/LAA/aortic thrombus, valvular vegetation, intracardiac tumor, aortic artery dissection, spontaneous ECHO contrast, PFO/ASA, and valvular abnormalities using prespecified data collection forms by one of the authors (S.S). We have presented the study characteristics, such as the first author's last name and the publication month and year in Table 1.

**Table 1.** Study characteristics and validity criteria.

| Study | Type of Study | >80% of Patients Received TEE | Tee Masked to Clinical Reviewer | Internal Validity [@] | External Validity (for Referral Center) [#] |
|---|---|---|---|---|---|
| | | **Criteria for Internal Validity** | | **Class [$]** | |
| Marino et al., 2016 [17] | Retrospective | Yes | No | II | Minor |
| Gaudron et al., 2014 [18] | Prospective | No | No | II | Minor |
| Zhang et al., 2012 * [19] | Retrospective | No | No | II | Minor |
| Knebel et al., 2009 [20] | Retrospective | Yes | No | II | Minor |
| de Bruijn et al., 2006 [6] | Prospective | Yes | No | II | Minor |
| Blum et al., 2004 [12] | Retrospective | Yes | No | II | Moderate |
| de Abreu et al., 2008 [21] | Prospective | No | No | II | Moderate |
| Harloff et al., 2006 [14] | Prospective | Yes | No | III | Minor |
| Shyu et al., 1993 * [22] | Prospective | Yes | No | III | Minor |
| Rauh et al., 1996 [23] | Prospective | Yes | No | III | Minor |
| Cujec et al., 1991 * [24] | Prospective | Yes | No | III | Minor |
| Pop et al., 1990 * [25] | Prospective | Yes | No | III | Minor |
| Pearson et al., 1991 * [26] | Prospective | Yes | No | III | Moderate |
| Censori et al., 1998 * [27] | Prospective | No | No | III | Moderate |
| Retting et al., 2008 [28] | Retrospective | Yes | No | III | Moderate |
| Total: 15 studies | | | | | |

* Patients/data were taken from the subgroup analysis only for patients meeting study criteria (without evidence of atrial fibrillation and/or heart disease and/or without indication for anticoagulation prior to TEE). [@] Internal validity was defined by three parameters (type of the study, if >80% of participants in the study received TEE, and if TEE was masked to clinical reviewer). [#] External validity was defined by the patient population enrolled in the study keeping the stroke referral center as standard. [$] The class of the study was determined based on the above-mentioned internal validity parameters.

### 2.7. Categories of Cardiac Abnormalities on TEE Findings

Patients with TEE findings of cardiac abnormalities were placed into four categories: (1) any cardiac abnormality reported on TEE, (2) finding of cardiac abnormalities that changed management at the time of the particular study and were noted in the respective studies, (3) findings of cardiac abnormalities that should change the management based on the current practice guidelines (this category includes LA/LAA/aortic thrombus, valvular vegetation, intracardiac tumor, and aortic arch dissection), (4) findings of cardiac abnormality might change management depending on the patient clinical situation and future research (this category includes patients with all abnormalities indicated in category 3 plus findings of PFO/ASA, valvular abnormalities, and spontaneous ECHO contrast) (Figure 2a).

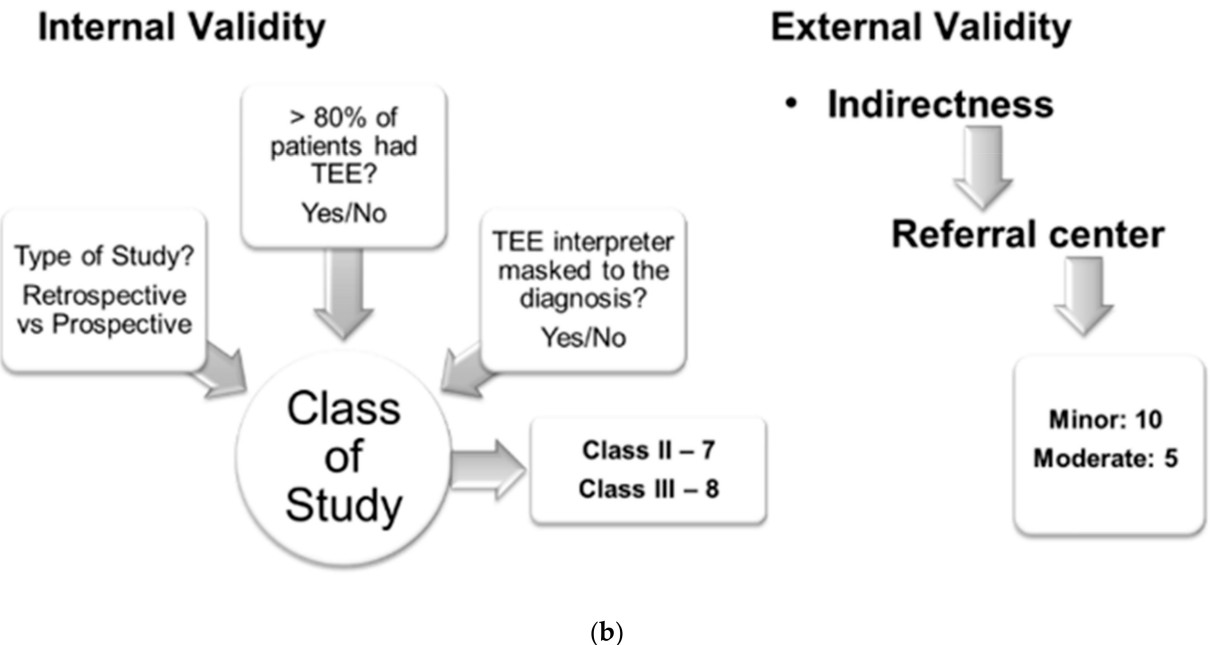

**Figure 2.** (**a**) Categories of cardiac abnormalities on TEE findings. (**b**) Internal and external validity evaluation of the included studies.

### 2.8. Statistical Analysis

A meta-analysis of proportion was performed using inverse variance random-effects models on the above-mentioned four categories. We also developed secondary spreadsheets capable of producing customized forest plots for evaluation of those additional findings and yields (the four categories mentioned above) of TEE. The prevalence of additional yields along with their 95% confidence intervals (95% CI) were mentioned in each study to evaluate cumulative yield. The risk of bias in prevalence studies was assessed using a modification of an existing tool and evidence of interrater agreement [29] (Figure S1). All studies were evaluated for internal and external validity by using the AAN

tool for evidence classification [30] (Figure 2b). The internal validity of study was accessed based on three parameters: (1) whether the study is prospective, (2) whether >80% of patients in the study received TEE, and (3) whether the TEE indication was masked to the clinical reviewer. If the study satisfies all three parameters with an answer 'yes' to the above-mentioned internal validity questions, then it is considered a class I study. With each 'no' answer to the questions, the class of a particular study will increase as internal validity decreases. With this criterion, we had a total of seven class II studies and eight class III studies. We did not find any class I studies. External validity was assessed based on the patient population in the studies considering a university-based referral center as the standard population. We found 10 studies with minor and 5 studies with moderate external validity (Table 1).

### 3. Results

Review of the PubMed database identified 524 potentially relevant articles, 492 duplicated articles, and irrelevant articles were excluded during the first round of review of titles only. During the second round, the abstracts of 32 articles were reviewed and 4 irrelevant articles were excluded. Full text reviews of the remaining 28 articles were conducted, of which 13 articles were excluded with reasons. Therefore, 15 studies were selected for final inclusion (Figure 1).

In our meta-analysis of 15 studies, a total of 2054 patients with CS or TIA were included who met the inclusion criteria. We found that 41.1% (845/2054) of patients had abnormal TEE findings but normal TTE (Table 2). The following TEE findings were noted across different studies: complex aortic plaques (349, 41.3%) was the most commonly reported finding followed by PFO (290, 34.3%), ASA (123, 14.5%), other valvular abnormalities (122, 14.4%), LA/LAA/aortic thrombus (90, 10.6%), PFO-ASA combination (62, 7.3%), spontaneous echo contrast (47, 5.6%), ASD (31, 3.7%), valvular vegetation (16, 1.9%), and intracardiac tumor (1, 0.1%) (Figure 3).

The meta-analysis of all 15 studies found that 37.5% (29.7%–45.1%) of patients had additional cardiac findings on TEE (Figure S2). The meta-analysis of 13 out of the 15 studies showed that 13.6% (8.1%–19.1%) of patients changed management after TEE evaluation (Figure S3). Our results also showed that, based on current guidelines, 4.1% (2.1%–6.2%) of patients should change management (Figure S4) and 30.4% (21.9%–38.9%) of patients could potentially change management after TEE evaluation (Figure S5) (Figure 4a).

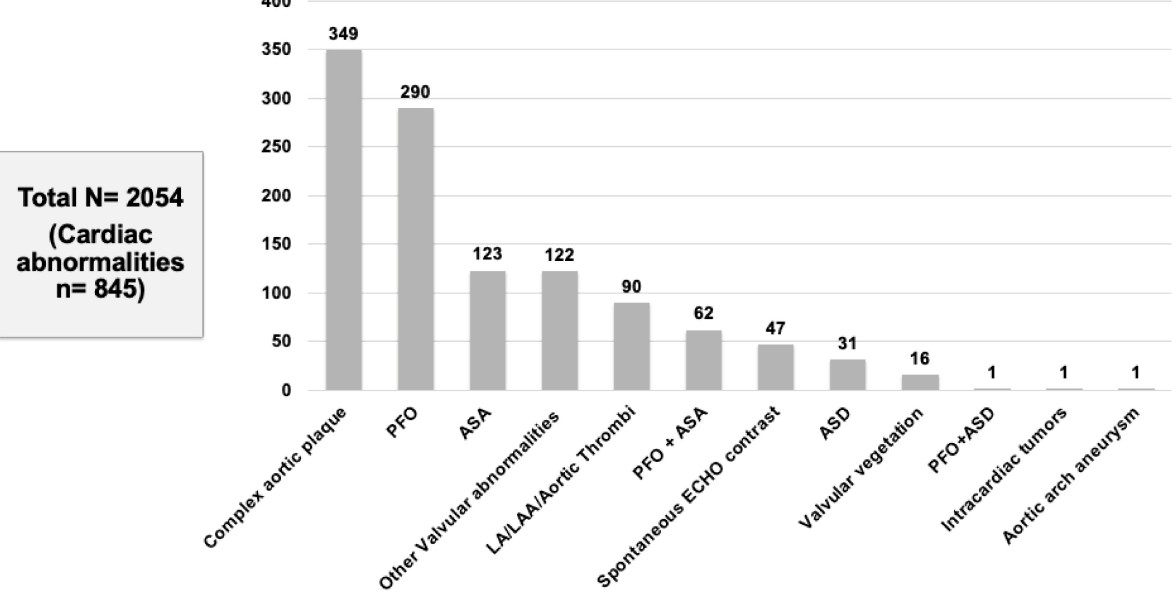

**Figure 3.** Cardiac abnormalities found on TEE.

**Table 2.** Studies showing cardiac abnormalities found on TEE that did change, should change, or could potentially change the management of CS/TIA patients after TEE evaluation.

| Study | Patients with TEE, Who Had Normal TTE | Additional TEE Findings not Reported on TTE | LA/LAA/Aortic Thrombi | Spontaneous ECHO Contrast | PFO | ASA | PFO + ASA | ASD | PFO + ASD | Complex Aortic Plaque | Intracardiac Tumors | Valvular Vegetation | Other Valvular Abnormalities | Aortic Arch Aneurysm | Required (Did) Change in the Management in the Study | Should Definitely Change Management per Current Guidelines @ | Could Potentially Change Management # |
|---|---|---|---|---|---|---|---|---|---|---|---|---|---|---|---|---|---|
| Marino et al., 2016 | 263 | 112 | 1 | 13 | 18 | 25 | 11 | 0 | 0 | 44 | 0 | 0 | 0 | NR | 1 | 1 | 68 |
| Gaudron et al., 2014 | 127 | 24 | 1 | 1 | 18 | 2 | 10 | 0 | 0 | 1 | 0 | 0 | 0 | NR | 12 | 1 | 32 |
| Zhang et al., 2012 * | 21 | 9 | 0 | 0 | 5 | 4 | 2 | 2 | 1 | 2 | 0 | 0 | 0 | NR | 6 | 0 | 11 |
| Knebel et al., 2009 | 702 | 369 | 17 | 18 | 152 | 51 | NR | 28 | NR | 102 | 1 | 14 | 111 | NR | NR | 32 | 364 |
| de Bruijn et al., 2006 | 231 | 90 | 38 | 3 | 9 | 3 | NR | NR | NR | 68 | NR | NR | 0 | NR | 38 | 38 | 53 |
| Blum et al., 2004 | 68 | 28 | 3 | NR | 5 | 0 | 0 | 1 | 0 | 23 | 0 | NR | NR | NR | 28 | 3 | 8 |
| de Abreu et al., 2008 | 84 | 27 | 7 | 1 | 3 | 10 | 2 | NR | NR | 23 | NR | NR | NR | NR | 27 | 7 | 23 |
| Harloff et al., 2005 | 212 | 65 | 14 | 5 | 43 | 8 | 31 | NR | NR | 37 | NR | NR | NR | NR | 17 | 14 | 101 |
| Shyu et al., 1993 * | 60 | 18 | 2 | 2 | NR | 7 | NR | NR | NR | 4 | NR | NR | 4 | NR | 18 | 2 | 15 |
| Rauth et al., 1996 | 30 | 24 | 3 | NR | 7 | 2 | 1 | NR | NR | 19 | NR | NR | NR | NR | 3 | 3 | 13 |
| Cujec et al., 1991 * | 39 | 7 | 1 | NR | 2 | 2 | NR | NR | NR | NR | 0 | 0 | 2 | NR | 1 | 1 | 7 |
| Pop et al., 1990 * | 53 | 26 | 1 | NR | NR | NR | NR | NR | NR | 22 | NR | 1 | 1 | 1 | 4 | 3 | 4 |
| Pearson et al., 1991 * | 38 | 7 | 0 | 0 | 2 | 2 | 4 | NR | NR | NR | NR | NR | 3 | NR | NR | 0 | 11 |
| Censori et al., 1998 * | 43 | 22 | 1 | 3 | 17 | 2 | NR | NR | NR | 2 | NR | NR | NR | NR | 1 | 1 | 23 |
| Retting et al., 2008 | 83 | 17 | 1 | 1 | 9 | 5 | 1 | NR | NR | 2 | NR | 1 | 1 | NR | 6 | 2 | 19 |
| Total: 15 studies | 2054 | 845 | 90 | 47 | 290 | 123 | 62 | 31 | 1 | 349 | 1 | 16 | 122 | 1 | 162 | 108 | 752 |
| | | | | 4 NR | 2 NR | 1 NR | 6 NR | 10 NR | 11 NR | 2 NR | 9 NR | 8 NR | 5 NR | 14 NR | 2 NR | | |

* Patients/data were taken from the subgroup analysis for patients meeting study criteria (without evidence of atrial fibrillation and/or heart disease and/or without indication for anticoagulation prior to TEE). @ Should definitely change in management per current guidelines: LA/LAA/aortic thrombus + valvular vegetation + intracardiac tumor + aortic artery dissection. # Could potentially change management: LA/LAA/aortic thrombus + valvular vegetation + intracardiac tumor + spontaneous ECHO contrast + PFO/ASA + valvular abnormalities. ASA: Atrial septal abnormality; ASD: atrial septal defect; LAA: left atrial appendage; PFO: patent foramen ovale; NR: not reported or measured in the study.

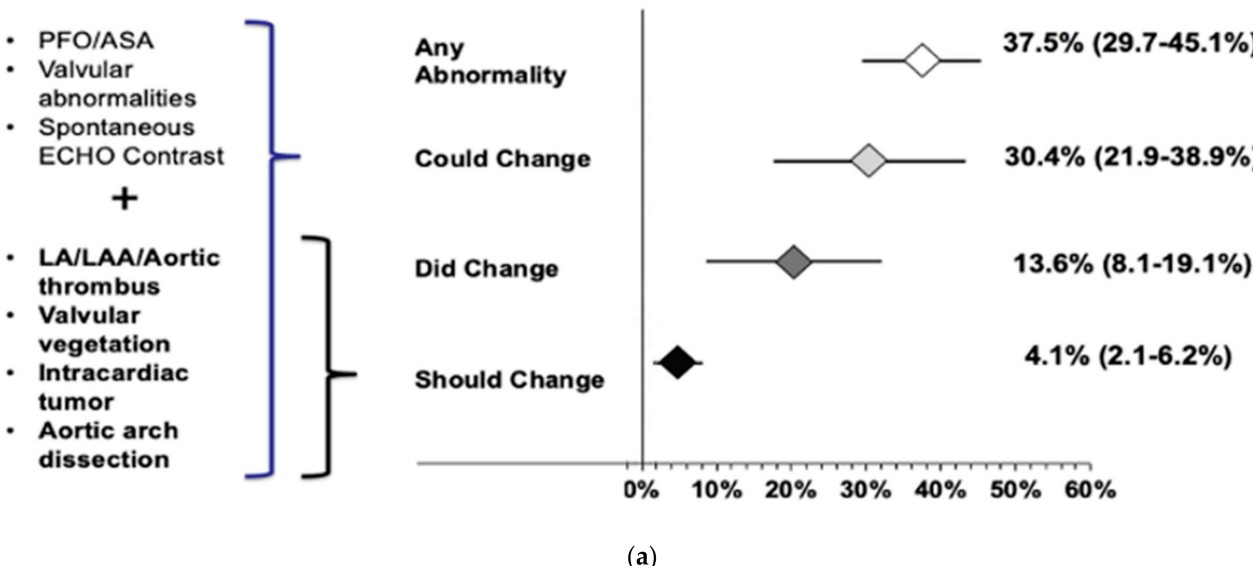

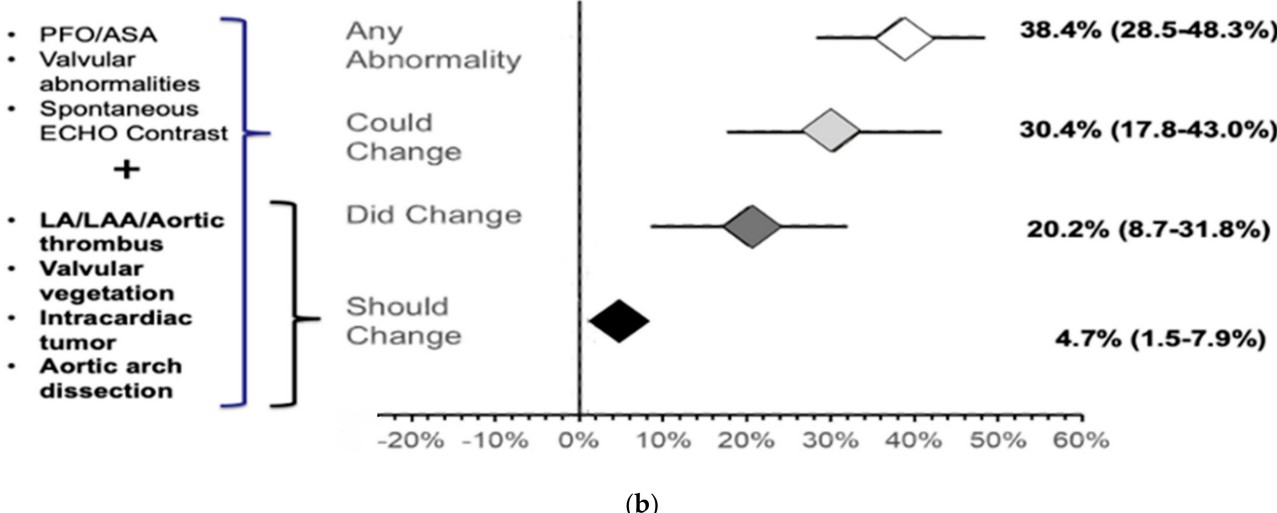

**Figure 4.** (**a**) Forest plot of additional cardiac abnormality on TEE evaluation and TEE findings that did change, should change, or could potentially change the management of CS/TIA patients after TEE evaluation. (**b**) Forest plot of additional cardiac abnormality on TEE evaluation and TEE findings that did change, should change, or could potentially change the management of CS/TIA patients after TEE evaluation in class II studies.

Subgroup analysis was also performed with only class II studies (7/15) to increase internal validity, which showed that 38.4% (28.5%–48.3%) of patients with CS and normal TTE are likely to have additional cardiac findings on TEE evaluation (Figure S6). The meta-analysis results of class II studies showed that TEE evaluation changed management in 20.2% (8.7%–31.8%) of patients (Figure S7) and should change management in 4.7% (1.5%–7.9%). Hence, the number needed to diagnose one additional abnormality with TEE that can change management per current guidelines is approximately 21 (Figure S8). Additionally, management could potentially be chanegd in 30.4% (17.8%–43.0%) (Figure S9) of patients, respectively (Figure 4b).

## 4. Discussion

Our systematic review and meta-analysis results show that the diagnostic yield of TEE to find additional cardiac findings in patients with cryptogenic stroke or TIA is approximately 38%, TEE could change management in 30.4% of patients, TEE has changed management in approximately 20% of these patients, and TTE should change management in approximately 4.7% of patients. The number needed to diagnose one additional abnormality with TEE that can change management per current guidelines is approximately 21.

Our study findings are similar to the systematic review and meta-analyses conducted by Katsanos et al. and McGrath et al. [31,32]. However, they did not perform internal and external validity analyses on their studies. Our study not only evaluated internal and external validity of the screened articles by using the AAN tool for evidence classification [30], but also provided a separate class II study analysis which increases the internal validity of the results. Our study also evaluated the number needed to diagnose one additional abnormality with TEE that can change management per current guidelines. In contrast to our study, Katsanos et al. 2016 did not look at the clinical impact of the abnormal cardiac findings; furthermore, they did not thoroughly exclude studies which did not perform or report normal TTE findings [31]. In comparison to our study, McGrath et al. 2014 did not include patients with cryptogenic TIAs in their study population [32].

Both of these studies (Katsanos et al. and McGrath et al.) reported that routine TEE in patients with cryptogenic stroke/TIA commonly identifies cardiac abnormalities [31,32]. They also reported that the variability in the prevalence of the common abnormal cardiac findings during TEE could be due to inter-study variation in the definition and prevalence of common cardiac abnormalities. Our study also reported variable prevalence of cardiac abnormalities on routine TEE consistent with the reported literature. In our study, the most prevalent cardiac finding reported during TEE was complex aortic plaques (41%), consistent with other studies which reported 51.2% aorta atherosclerosis [31], 21.8% complex descending aorta atheromatosis [11], and 16.7% complex plaque of the ascending aorta or arch [17] in cryptogenic stroke/TIA patients undergoing TEE. Furthermore, the second most prevalent finding of PFO (34%) followed by ASA (14.5%) was consistent with other studies [11,17,31]. PFO in patients with CS or TIA are thought to be more frequently associated with ASAs compared with the PFOs in asymptomatic patients [33]. Another meta-analysis by Kasanos et al. reported that medically treated PFO patients do not have a higher risk of recurrent CS compared to patients without PFO and they are also unrelated to size of the PFO [34].

The clinical impact of the findings from TEE has been an aggressive topic in recent years and appropriate management of a few findings is still questionable considering the heterogeneous stroke populations in the majority of the studies. We found that TEE has changed management in approximately 20% of patients with CS/TIA and should change management in 4.7% of the patients as per guidelines. In support of our findings, a study by Harloff et al. reported that 30.7% of CS patients were started on oral anticoagulants due to TEE findings primarily for PFO, ASA, or aortic plaques < 4 mm [14]. The TEE findings of intracardiac thrombus, or tumor and infective endocarditis, would warrant anticoagulation therapy or surgery as per current American Heart Association/American Stroke Association guidelines [35,36]. However, these findings are uncommon, and our meta-analysis found valvular vegetations in only 2% of patients and intracardiac tumors in 0.1% of patients. In support of our study's meta-analysis, Katsanos et al. [34] also found left atrial thrombus in 3% of patients and intracardiac tumors in 0.2% of patients.

In our study, we did not include PFO findings in the 'should change management' category; rather, it was added in the 'could change management' category. Based on recent meta-analysis of PFO closure trials, PFO closure in selected patients with cryptogenic stroke/TIA can help prevent recurrent strokes compared to medical management alone [37]. Our study was unable to identify patients meeting criteria for PFO closure post TEE across different cohort studies, which might underestimate the calculated yield of TEE to change clinical management in that particular category.

Apart from evolving guidelines, the rate of TEE affecting clinical management varies between individual stroke providers, and there are no data reporting the impact of these management decisions on patients' outcomes. In addition, complex aortic plaque is a common TEE finding, but it is difficult to determine its impact on clinical management due to a lack of evidence-based guidelines for evaluating and treating proximal aorta atherosclerosis and variability among different practices.

*Strength and Limitations*

To our knowledge, ours is the one of the few large studies that shows a diagnostic yield of TEE in cryptogenic stroke and TIA patients, using a meta-analysis of 2054 patients. Compared to previously published studies, our study provides a separate class II study analysis which improves the internal validity of the results. It also finds a number needed to diagnose one additional abnormality with TEE which can impact clinical outcome. Our findings are limited due to the unavailability of risk-stratification in our meta-analysis. A large observational study would probably be able to enlighten the relationships between multiple risk factors and reduce the risk of ecological fallacy while attempting to make inferences about individuals using study-level information. In addition to that, our study findings should be further evaluated in future studies for their external validity. One other limitation of our meta-analysis is selection bias across the various studies, which we have addressed using criteria for internal and external validity.

## 5. Conclusions

This study not only focused on the diagnostic yield of TEE in cryptogenic stroke and TIA patients with normal TTE, but also assessed the clinical impact of these additional cardiac findings. Based on this meta-analysis, TEE can find additional cardiac abnormalities in approximately 38% of the studied patients. Out of which, clinical management should be altered in approximately 4.7% of patients. That translates into the number needed to diagnose one additional abnormality with TEE in a given population which should change management (approximately 21). Clinicians should consider obtaining TEE in cryptogenic stroke and TIA patients with normal TTE findings.

**Supplementary Materials:** The following are available online at https://www.mdpi.com/article/10.3390/neurolint13040063/s1, Figure S1: The risk of bias assessment; Figure S2: Forest plot of proportion of additional cardiac findings on TEE; Figure S3: Forest plot of proportion of cardiac abnormalities that did change management; Figure S4: Forest plot of proportion of cardiac abnormalities that should change management; Figure S5: Forest plot of proportion of cardiac abnormalities that could change management; Figure S6: Forest plot of proportion of additional cardiac findings on TEE in class II studies Figure S7: Forest plot of proportion of cardiac abnormalities that did change management in class II studies; Figure S8: Forest plot of proportion of cardiac abnormalities that should change management in class II studies; Figure S9: Forest plot of proportion of cardiac abnormalities that could change management in class II study.

**Author Contributions:** Conceptualization, S.S., Y.W. and G.S.G.; methodology, G.S.G.; software, G.S.G.; validation, S.S., P.M. and U.P.; formal analysis, G.S.G.; investigation, S.S. and Y.W.; resources, S.S., P.M. and U.P.; data curation, S.S., P.M. and U.P.; writing—original draft preparation, S.S., P.M. and U.P.; writing—review and editing, Y.W. and G.S.G.; visualization, S.S.; supervision, Y.W. and G.S.G.; project administration, S.S. All authors have read and agreed to the published version of the manuscript.

**Funding:** No funding was needed for this study.

**Institutional Review Board Statement:** All procedures performed in this study were in accordance with the ethical standards of the institutional and/or national research committee and with the 1964 Helsinki Declaration and its later amendments or comparable ethical standards. All the patients' related information were de-identified before publishing.

**Informed Consent Statement:** Not applicable.

**Acknowledgments:** Not applicable.

**Conflicts of Interest:** The authors declare no conflict of interest.

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
