# Peer review of "Diagnostic Yield of TEE in Patients with Cryptogenic Stroke and TIA with Normal TTE: A Systematic Review and Meta-Analysis"

_2035-8377, doi:10.3390/neurolint13040063_

Round 1

Reviewer 1 Report

It is an interesting study about review about transesophageal echocardiography (TEE) with normal transthoracic echocardiography. There were 37.5% additional cardiac findings on TEE and could potentially change management in 30.4% of patients.

The embolic stroke was also indicated to perform TEE. There were complex aortic plaques(41.3%) were the most commonly reported findings followed by PFO (34.3%), ASA (14.5%), other valvular abnormalities ( 14.4%), LA/LAA/Aortic thrombus (10.6%). The findings were similar with the previous study (reference 19). Please discuss the strength of your review compared to the previous study. There was some risk for TEE performance, how to encourage the patients to receive this risky study?

Author Response

Dear Reviewer 1

I appreciate the time and effort you have put into making the manuscript accurate and reader-friendly.

Attached is the explanation for your comments 

Please discuss the strength of your review compared to the previous study. 

We included fifteen studies with 2054 patients, in comparison to Katsanos et al., our study has the following additional findings:

  • We excluded studies that did not perform or reported normal TTE findings before getting TEE
  • In our study, sensitivity analysis was also performed with class II studies to increase internal validity
  • Our secondary aim was evaluated by the number needed to diagnose one additional abnormality with TEE that can change management per current guidelines

There was some risk for TEE performance, how to encourage the patients to receive this risky study?

As this was a systematic review and meta-analysis, we do not need to encourage any patient to be part of this study. All the studies we have included were observational studies and NOT RCTs so none of them requires patients to convince patients to underwent for TTE. All observational studies just collect the data from EHR without changing the management on basis of the study. 

Feel free to ask us any other concerns you may have.

Thank you

Reviewer 2 Report

Through this systematic review, it is inferred that the authors have intended to stress excellence based on TEE diagnosis in identifying the new diseases related to CS/TIA symptoms. However, it has not been emphasized at all such an intention of the authors in the composition of the overall content. Moreover, it has not shown whether such a report has any clinical implications and whether it can be provided any benefit. From a broad perspective, the following issues need to be addressed: 1. Is the ultimate goal to emphasize the diagnostic excellence of TEE, or? For your information, the authors are only raising problems of the absence of definite guidelines for TEE and TTE. That is, the clear rationale for any problems related to the TEE diagnosis in clinical did not provide. 2. Through the existing reports, it has been well known that the diagnostic method of TEE even for CS and TIA patients is excellent in many ways. If so, it is wondering what kind of discrimination of this review can be claimed compared with the existing reports. 3. In conclusion, the author should be able to clearly state what issues could have addressed through this investigation. However, because the clinical rationale for this investigation is unclear, there is concerned that the conclusions may not be considered in line with the logical flow.

Author Response

Dear Reviewer-2,

I appreciate the time and effort you have put into making the manuscript accurate and reader friendly.

Attached is the explanation for your comments 

From a broad perspective, the following issues need to be addressed:

  1. Is the ultimate goal to emphasize the diagnostic excellence of TEE, or? For your information, the authors are only raising problems of the absence of definite guidelines for TEE and TTE. That is, the clear rationale for any problems related to the TEE diagnosis in clinical did not provide.

- Thank you and appreciate reviewer’s comments in this matter. Based on the current literature and guidelines, the utility of obtaining TEE in cryptogenic stroke and TIA is unclear. With this meta-analysis, we have calculated additional benefit of obtaining TEE in this specific population rather than focusing on the diagnostic excellence of TEE.

  1. Through the existing reports, it has been well known that the diagnostic method of TEE even for CS and TIA patients is excellent in many ways. If so, it is wondering what kind of discrimination of this review can be claimed compared with the existing reports.

- Appreciate reviewer’s comment. It has been established that, overall, it is a better study compared to TTE, however it is not well established how much beneficial it is in this specific patient population (cryptogenic strokes and TIA). Are there any additional benefits of obtaining TEE after normal TEE? What is the additional yield? Should clinicians consider this study? Obtaining additional study with TEE will help change management of these patient population? This study aims to answer all these questions.  It is mentioned in the last paragraph of the introduction section.

  1. In conclusion, the author should be able to clearly state what issues could have addressed through this investigation. However, because the clinical rationale for this investigation is unclear, there is concerned that the conclusions may not be considered in line with the logical flow.

- Thank you for the comment. Conclusion has been updated per reviewer’s recommendations. (Line 256-260) It clearly emphasize number needed to diagnoses/ clinical yield.

Feel free to ask us any other concerns you may have.

Thank you

Round 2

Reviewer 1 Report

The cryptogenic cerebral ischemia was an indication for the transesophageal echocardiography (TEE) study. There was a similar study that showed the same findings (European Journal of Neurology 2016, 23: 569–579). It seems no new idea for the readers.

The TEE must study for cryptogenic cerebral ischemia or transient ischemic stroke, not change transthoracic echocardiography to TEE. It is difficult to understand the goal of this study to emphasize and new suggestions for the readers.

Author Response

Dear reviewer, 

Thank you for your time to review this article. We agree with your comments and here is our response. We have also made changes to the article based on your comments. 

The cryptogenic cerebral ischemia was an indication for the transesophageal echocardiography (TEE) study. There was a similar study that showed the same findings (European Journal of Neurology 2016, 23: 569–579). It seems no new idea for the readers.

            Thank you for bringing up this study again. The reviewer is referring to the same study, which was mentioned in round 1, and the authors provided a detailed report. This study is Katsanos et al. 2016, which is reference 19 in our study.

We would like to thoroughly readdress the significant difference between our study and Katsanos et al. 2016 (or European Journal of Neurology 2016, 23: 569–579 – mentioned in round 2 again by reviewer 1).

            Compared to Kantsanos et al. 2016, our study, not only evaluated the internal and external validity of the screened articles by using the AAN tool for evidence classification (18), but also provided a separate Class II study analysis which increases the internal validity of the results. Our study also evaluated the number needed to diagnose one additional abnormality with TEE that can change management per current guidelines. In contrast to our study, Katsanos et al. 2016 did not look at the clinical impact of the abnormal cardiac findings, furthermore it did not thoroughly exclude studies which did not perform or report normal TTE findings.

            As mentioned above, we believe there is a significant methodological difference between our study and Kantsanos et al. 2016. We have now thoroughly included this explanation in the discussion section as per reviewers’ comments (line 199-208).

The TEE must study for cryptogenic cerebral ischemia or transient ischemic stroke, not change transthoracic echocardiography to TEE. It is difficult to understand the goal of this study to emphasize and new suggestions for the readers.

            Thank you for the comment. Our study does not suggest or imply in any way to substitute transthoracic echocardiography (TTE) with transesophageal echocardiography (TEE). There are no guidelines on when to pursue TEE in cryptogenic stroke or TIA patients and how much it can impact clinical management. Our study was performed to answer this question by identifying the additional yield or impact of obtaining TEE in such patient population with normal TTE. This study will certainly help clinicians to make an informed decision regarding further workup in this patient population to further identify cardiac etiology and subsequently optimize secondary stroke prevention if finds an appropriate etiology.

Reviewer 2 Report

All concerns have been well addressed.

Author Response

Dear Reviewer,

I appreciate your time and effort to make this article more accurate and reader friendly.

I am glad that we can able to address all concerns.

Feel free to contact us if you have any concerns.

Thank you.